# Simultaneous Expression of Th1- and Treg-Associated Chemokine Genes and CD4^+^, CD8^+^, and Foxp3^+^ Cells in the Premalignant Lesions of 4NQO-Induced Mouse Tongue Tumorigenesis

**DOI:** 10.3390/cancers13081835

**Published:** 2021-04-12

**Authors:** Hana Yamaguchi, Miki Hiroi, Kazumasa Mori, Ryosuke Ushio, Ari Matsumoto, Nobuharu Yamamoto, Jun Shimada, Yoshihiro Ohmori

**Affiliations:** 1Division of Microbiology and Immunology, Department of Oral Biology and Tissue Engineering, Meikai University School of Dentistry, 1-1 Keyakidai, Sakado, Saitama 350-0283, Japan; hana@dent.meikai.ac.jp (H.Y.); mikih@dent.meikai.ac.jp (M.H.); r-ushio-007@dent.meikai.ac.jp (R.U.); matsuari@dent.meikai.ac.jp (A.M.); 2First Division of Oral and Maxillofacial Surgery, Department of Diagnostic and Therapeutic Sciences, Meikai University School of Dentistry, 1-1 Keyakidai, Sakado, Saitama 350-0283, Japan; kazu-mori@dent.meikai.ac.jp (K.M.); nyamamot@dent.meikai.ac.jp (N.Y.); shimadajun@mac.com (J.S.)

**Keywords:** chemokine, Th1, Th2, Treg, Foxp3, gene expression, oral squamous cell carcinoma, premalignant, carcinogen, 4-nitroquinoline 1-oxide

## Abstract

**Simple Summary:**

Oral squamous cell carcinoma (OSCC), the most common oral malignancy, severely impacts patient quality of life because of oro-facial destruction. OSCC is preceded by oral premalignant lesions (OPLs). Moreover, lower T cell infiltration in OPLs is associated with OSCC, suggesting that T cell-mediated adaptive immunity protects against malignant transformation. In this study, we used the carcinogen 4NQO, which mimics tobacco-related carcinogenesis, in a mouse model to examine the gene expression kinetics of chemokines/cytokines during OPL and OSCC development. Our results demonstrate that both Th1- and Treg-associated chemokines were simultaneously expressed in 4NQO-induced OPL, with their expression correlating with the infiltration of CD8^+^ and Foxp3^+^ cells, respectively. These results indicate that antitumor immune responses and immunosuppression are simultaneously initiated during OLP development.

**Abstract:**

Chemokines and cytokines in the tumor microenvironment influence immune cell infiltration and activation. To elucidate their role in immune cell recruitment during oral cancer development, we generated a mouse tongue cancer model using the carcinogen 4-nitroquinoline 1-oxide (4NQO) and investigated the carcinogenetic process and chemokine/cytokine gene expression kinetics in the mouse tongue. C57/BL6 mice were administered 4NQO in drinking water, after which tongues were dissected at 16 and 28 weeks and subjected to analysis using the RT^2^ Profiler PCR Array, qRT-PCR, and pathologic and immunohistochemical analyses. We found that Th1-associated chemokine/cytokine (*Cxcl9*, *Cxcl10*, *Ccl5*, and *Ifng*) and Treg-associated chemokine/cytokine (*Ccl17*, *Ccl22*, and *Il10*) mRNA levels were simultaneously increased in premalignant lesions of 4NQO-treated mice at 16 weeks. Additionally, although levels of *Gata3*, a Th2 marker, were not upregulated, those of *Cxcr3*, *Ccr4*, and *Foxp3* were upregulated in the tongue tissue. Furthermore, immunohistochemical analysis confirmed the infiltration of CD4^+^, CD8^+^, and Foxp3^+^ cells in the tongue tissue of 4NQO-treated mice, as well as significant correlations between Th1- or Treg-associated chemokine/cytokine mRNA expression and T cell infiltration. These results indicate that CD4^+^, CD8^+^, and Foxp3^+^ cells were simultaneously recruited through the expression of Th1- and Treg-associated chemokines in premalignant lesions of 4NQO-induced mouse tongue tissue.

## 1. Introduction

Oral squamous cell carcinoma (OSCC), the most common oral malignancy, severely impacts patient quality of life due to tumor invasion, orofacial destruction, and metastasis [1]. Despite recent advances in the understanding and management of OSCC, its five-year survival rate remains low (~50–60%) [2]. Although OSCC is preceded by clinically evident oral premalignant lesions (OPLs) of leukoplakia, characterized by a circumscribed thickening of the mucosa covered by whitish patches with epithelial dysplasia [3], not all OPLs develop into OSCC, and some may even regress [4]. Recent immunohistochemical (IHC) and bioinformatics analyses have shown that lower T cell infiltration in OPLs is associated with OSCC [5,6,7], suggesting that T cell-mediated adaptive immunity protects against malignant transformation. Our previous study on IHC analysis of OPLs showed that the tumor microenvironment is a Th1-dominated milieu, where cells expressing chemokine receptor CXCR3 and CCR5, typically expressed on CD4^+^ T helper type 1 (Th1), CD8^+^ cytotoxic T lymphocytes (CTLs), and natural killer (NK) cells, are observed in OPLs [8]. Although T cell infiltration in the tumor microenvironment is a key component to protecting against malignant transformation, the kinetics of chemokine expression in the progression of premalignant lesions to OSCC remain to be fully elucidated.

The water-soluble chemical carcinogen 4-nitroquinoline 1-oxide (4NQO) forms DNA adducts, resulting in adenosine-to-guanosine substitution [9,10,11]. Additionally, 4NQO induces intracellular oxidative stress, which generates reactive oxygen species that induce mutations and DNA-strand breaks. These carcinogenic properties are similar to the genetic and molecular alterations induced by tobacco carcinogens [10]. Further, 4NQO administration in mice and rats through drinking water results in pathohistological characteristics similar to those of human oral carcinogenesis from premalignant lesions through cancerous lesions [9,12]. Therefore, 4NQO-induced oral cancer is an excellent model for investigating the kinetics of chemokines and the role of infiltrating immune cells during the development and progression of oral carcinogenesis.

In this study, we investigated the expression of chemokine genes in 4NQO-induced mouse tongue carcinogenesis. The results demonstrate that genes for both Th1- and Treg-associated chemokines were expressed simultaneously in 4NQO-induced murine OPLs, and that, as the tumor progressed to OSCC, the expression of Th1- and Treg-associated chemokine genes declined. IHC analysis revealed significant positive correlations between the infiltration of CD4^+^ or CD8^+^ cells and the expression of Th1-associated chemokine genes, as well as between the infiltration of Foxp3^+^ cells and Treg-associated chemokine genes. These results indicate that CD4^+^, CD8^+^, and Foxp3^+^ cells were simultaneously recruited through the expression of Th1- and Treg-associated chemokines in the premalignant lesions of carcinogen-induced mouse tongue tissues.

## 2. Materials and Methods

### 2.1. Animals and Carcinogen Treatment

Female C57BL/6 mice (4–10 weeks old) purchased from the Jackson Laboratory (Bar Harbor, ME, USA) were used for this study. Briefly, freshly prepared 4NQO (Sigma-Aldrich, St. Louis, MO, USA) stock solution (2.5 mg/mL in ethanol) was added to the drinking water at 50 μg/mL, and the water was changed once weekly. Mice were randomly divided into an experimental group provided with drinking water containing 4NQO (*n* = 12) and into a control group (vehicle) provided with drinking water without 4NQO (*n* = 12) but the same volume of ethanol. Mice were allowed access to drinking water at all times during the treatment. After 16 weeks of 4NQO treatment, the mice were provided regular water (without carcinogen) for another 12 weeks. The use of 50 μg/mL 4NQO to induce premalignant and cancerous lesions in the tongue has been previously reported [13,14]. Mice were sacrificed at 16 or 28 weeks, and the tongue lesions were assessed by gross and histological examination.

Whole tongue tissues were removed, macroscopic images were obtained, and the tongues were cut lengthwise into halves. One half was temporarily stored in RNAlater (Ambion, Austin, TX, USA) at −80 °C for extraction of total RNA, and the other half was fixed in 4% paraformaldehyde and embedded in paraffin. This study was approved by the Institutional Laboratory Animal Care guidelines of the Meikai University School of Dentistry on 15 March 2016 (approval code: A1636).

### 2.2. Pathologic Examination

After sectioning paraffin-embedded tissues, they were deparaffinized, rehydrated, and stained with hematoxylin and eosin (H-E). Squamous neoplasia was histologically evaluated based on the World Health Organization classification of head and neck tumors [15]. The lesions were classified into four grades: hyperplasia, mild dysplasia, moderate dysplasia, and carcinoma. Hyperplasia was defined as increased cell numbers in the spinous layer and/or basal layers without cellular atypia. Mild dysplasia was defined as loss of polarity in the lower third of the epithelium accompanied by cytological atypia. Architectural disturbance of the polarity extending into the middle third of the epithelium was defined as moderate dysplasia. Carcinoma was defined when a full-thickness architectural abnormality in the viable cellular layer was accompanied by pronounced cytologic atypia, disturbance of the keratin layer, and invasion through the basement membrane.

### 2.3. Preparation of Total RNA and the Polymerase Chain Reaction (PCR) Array

Total RNA was extracted from the tissue samples using the RNeasy Fibrous Tissue Mini Kit (Qiagen, Hilden, Germany), and 0.5 µg of each total RNA sample were used for reverse transcription in a 20 μL reaction using the RT^2^ First Strand Kit (Qiagen). The prepared cDNA was then diluted with 91 μL of nuclease-free water and added to the RT^2^ qPCR SYBR Green Master Mix (Qiagen) according to the protocol for the RT^2^ Profiler PCR Array. Twenty-five microliters of the PCR cocktail were then added to each well of the RT^2^ Profiler PCR Array (PAMM-150Z, Qiagen) in 96-well plates to detect the expression of 84 genes encoding mouse cytokines and chemokines and five housekeeping genes (*Actb*, *B2m*, *Gapdh*, *Gusb*, and *Hsp90ab1*). The gene lists for the RT^2^ Profiler PCR Array are shown in Appendix A. The PCR mixture was amplified using a LightCycler 480 Real-Time PCR System (Roche Diagnostics, Basel, Switzerland) and the following PCR cycling conditions: 95 °C for 10 min, followed by 40 cycles of 95 °C for 15 s, 60 °C for 1 min, and 55 °C for 30 s. PCR data were analyzed using the PCR Array Data Analysis Web portal (Qiagen), and gene expression levels were normalized against five housekeeping genes. Fold changes in gene expression were calculated using the ΔΔCt method. A change of ≥ 2-fold in gene expression was considered significant (*p* < 0.05) based on a web portal analysis.

### 2.4. Quantitative Reverse Transcription (qRT)-PCR

Each total RNA sample (0.5 µg) was used for reverse transcription in a 20 μL reaction using a high-capacity cDNA reverse transcription kit (Life Technologies, Carlsbad, CA, USA). cDNA (1 µL) was then amplified using a LightCycler 480 Real-Time PCR System (Roche Diagnostics) and TaqMan Gene Expression Master Mix (Life Technologies, Carlsbad, CA, USA) according to manufacturer instructions. Real-time PCR probes and primers (Appendix A) were selected using the Universal Probe Library Assay Design Center (Roche Diagnostics). The PCR cycling conditions were as follows: 95 °C for 10 min, followed by 40 cycles of 95 °C for 15 s, 60 °C for 1 min, and 50 °C for 30 s. PCR data were analyzed using LightCycler 480 quantification software (Roche Diagnostics), and transcript levels were calculated relative to the 18S rRNA levels as an internal control using the E-methods with standard-curve-derived efficiencies.

### 2.5. IHC Staining

The tissue sections were deparaffinized, immersed in 10 mM citrate buffer (pH 6.0), and heated in a microwave for 15 min for antigen retrieval. After rinsing in phosphate-buffered saline (PBS), the sections were incubated with 3% hydrogen peroxide in methanol for 10 min to block endogenous peroxidase activity. Endogenous avidin and biotin were blocked using the Avidin/Biotin Blocking Kit (Zymed Laboratories, San Francisco, CA, USA) at 25 °C for 10 min. To reduce nonspecific antibody binding, the samples were exposed to 2% bovine serum albumin for 30 min. The tissue sections were then incubated with rabbit monoclonal anti-mouse CD4 antibody (1:100, Cat#25229, Cell Signaling Technology, Danvers, MA, USA), rabbit monoclonal anti-mouse CD8a antibody (1:200, Cat#98941, Cell Signaling Technology), or rabbit monoclonal anti-mouse Foxp3 antibody (1:100, Cat#12653, Cell Signaling Technology) at 1:100 and 25 °C in a humidified chamber for 60 min. The tissue sections were then washed in PBS and incubated with horseradish peroxidase (HRP)-labeled anti-rabbit antibodies (Dako EnVision System; HRP-labeled polymer; Dako, Kyoto, Japan) for 30 min. Peroxidase activity was visualized by immersing the tissue sections in Super Sensitive DAB (BioGenex, Fremont, CA, USA). Finally, the tissue sections were counterstained with Mayer’s hematoxylin and mounted. The immunostained tissue sections were imaged using a virtual slide scanner (NanoZoomer-XR; Hamamatsu Photonics, Hamamatsu, Japan), which was outsourced to the New HistoScience Laboratory (Oume, Japan). The number of positively stained cells was counted in the entire section of each specimen under high-power magnification (400×).

### 2.6. Statistical Analyses

One-way analysis of variance for multiple data was used to test for statistically significant differences in Prism 9 software (Version 9.1.0, GraphPad Software, San Diego, CA, USA). Correlations between different types of chemokine gene expression or infiltrated immune cells were tested using nonparametric Spearman’s rank analysis. A two-sided *p* < 0.05 was considered statistically significant.

## 3. Results

### 3.1. Development of Premalignant and Carcinoma Tongue Lesions in 4NQO-Treated Mice

To elucidate the role of chemokines/cytokines in oral cancer development, we used a mouse tongue cancer model using the carcinogen 4NQO and investigated the carcinogenesis process and chemokine/cytokine gene expression kinetics in the mouse tongue. Mice were treated with 4NQO in drinking water for 16 weeks and then changed to regular drinking water for another 12 weeks. Although no noticeable gross lesions were observed in the vehicle groups (Figure 1A), gross lesions became increasingly evident at 5–6 weeks after 4NQO treatment and developed over the following 6–12 weeks (Figure 1B). Numerous whitish exophytic patches were observed at 28 weeks (Figure 1C). Histopathologic analysis of tissue specimens from 4NQO-treated mice (Figure 1E,F) showed that premalignant lesions (hyperplasia and mild dysplasia) were observed at 16 weeks (Figure 1G), with more severe pathological grades (moderate dysplasia and SCC) evident at 28 weeks (Figure 1H). No obvious histopathological alterations were observed in any of the vehicle groups (Figure 1D).

### 3.2. PCR Array Analysis of Mouse Chemokine/Cytokine Gene Expression in Tongue Tissues from 4NQO-Treated Mice

We initially examined chemokine/cytokine gene expression profiles during the initiation and development of 4NQO-induced tongue tumors using PCR array analysis. Of the 84 genes analyzed, 21 were upregulated by > 2-fold in 4NQO-treated mice at 16 weeks (Figure 2A), and 33 genes were upregulated at 28 weeks after 4NQO treatment (Figure 2B). The expression of *Ccl20, Cxcl10, Ccl22, Il1b*, and *Cxcl1* was increased by > 10-fold at 16 weeks. Interestingly, *Cxcl10* and *Ccl22*, which are Th1- and Th2/Treg-associated chemokines, respectively [16,17], were upregulated in the early stages of mouse tongue tumorigenesis. CXCL10 recruits CD4^+^ Th1, CD8^+^ CTLs, and NK cells, whereas CCL22 is a ligand for the chemokine receptor CCR4, expressed on CD4^+^ Treg cells, as well as on CD4^+^ Th2 cells [17]. Therefore, we further validated the expression of Th1-associated chemokine genes, including *Cxcl9*, *Cxcl10*, and *Ccl5*, and Th2/Treg-associated chemokine genes, including *Ccl17* and *Ccl22*, in carcinogen-induced tongue tumors using qRT-PCR.

### 3.3. Th1-Associated Chemokine Gene Expression in 4NQO-Induced Mouse Tongue Tumorigenesis

Transcript levels of Th1-associated chemokines, such as *Cxcl9*, *Cxcl10*, and *Ccl5*, were upregulated in the tongue tissues of 4NQO-treated mice at 16 weeks, and the upregulated expression was maintained for up to 28 weeks (Figure 3A–C). Strong positive correlations were observed between the transcript levels of *Cxcl10* and *Cxcl9* (*p* < 0.0001), *Ccl5* (*p* = 0.0003), and *Ifng* (*p* < 0.0001) (Figure 3D–F). Consistent with the upregulation of Th1-associated chemokine gene expression, the transcript level of *Ifng* was also upregulated in the tongue tissue of 4NQO-treated mice (Figure 3G), with similar positive correlations observed between *Ifng*, *Cxcl9*, and *Ccl5* (Figure 3H,I).

As CXCL9 and CXCL10 are ligands for chemokine receptor CXCR3, we examined the transcript levels of *Cxcr3* in the tongue tissues of 4NQO-treated mice (Figure 4). *Cxcr3* transcripts were significantly upregulated in the tongue tissues at 16 weeks after 4NQO-treatment, and the elevated transcript levels were decreased at 28 weeks (Figure 4A). Additionally, *Cxcr3* transcript level was positively correlated with *Cxcl9*, *Cxcl10,* and *Ifng* levels (*p* < 0.0001; Figure 4B–D). Pathologic examination showed that premalignant dysplasia and SCC lesions were observed in the 4NQO-treated mice at 28 weeks (Figure 1H); therefore, we examined whether levels of Th1-associated chemokines were altered in dysplasia and SCC lesions (Figure 5). The transcript levels of Th1-associated chemokines/cytokines (*Cxcl9*, *Cxcl10*, *Ccl5*, and *Ifng*) and the Th1-associated chemokine receptor *Cxcr3* were upregulated in dysplasia, whereas the expression of these Th1-associated genes was decreased in SCC. These results suggest that Th1-associated chemokines recruit IFNγ-producing CXCR3^+^ cells in premalignant lesions during 4NQO-induced tumorigenesis, and, as the tumor progresses to SCC, expression of the Th1-associated chemokines/cytokines decline.

### 3.4 Th2/Treg-Associated Chemokine Gene Expression in 4NQO-Induced Mouse Tongue Tumorigenesis

Transcript levels of Th2/Treg-associated chemokines *Ccl17* and *Ccl22* were also upregulated in the tongue tissues of 4NQO-treated mice at 16 weeks (Figure 6A,B); however, their elevated expression was decreased at 28 weeks. Additionally, we observed a strong positive correlation between transcript levels of *Ccl17* and *Ccl22* (*p* < 0.0001; Figure 6C). To determine whether transcripts of Th2/Treg-associated cytokines were detected in tumor lesions, we examined mRNA levels of *Il4, Il5*, and *I10* in tongue tissues from 4NQO-treated mice. Although transcript levels of *Il4* and *Il5*, which are signature cytokines produced during type 2 immune responses [18], were not upregulated in 4NQO-treated mice (Appendix A), *Il10* level was significantly upregulated in tongue tissues from 4NQO-treated mice at 16 weeks (Figure 6D). Moreover, Spearman’s rank correlation coefficient analysis demonstrated a strong positive correlation between transcript levels for *Ccl17* and *Il10* (*p* = 0.0002; Figure 6E) and a moderate positive correlation between *Ccl22* and *Il10* (*p* = 0.0058; Figure 6F).

As CCL17 and CCL22 are ligands for the chemokine receptor CCR4, we examined transcript levels of *Ccr4*, expressed on Th2 cells and Tregs [16], in tongue tissues from 4NQO-treated mice (Figure 7A). We found that *Ccr4* level was upregulated in tongue tissues at 16 weeks after 4NQO treatment and decreased at 28 weeks. Additionally, we observed positive correlations between *Ccr4* levels and those of Th2/Treg-associated chemokines *Ccl17* (*p* = 0.0076; Figure 7B), *Ccl22* (*p* = 0.0012; Figure 7C), and *Il10* (*p* = 0.0017; Figure 7D).

To determine whether the elevated *Ccr4* expression derived from Th2 cells or Tregs, we examined transcript levels of *Gata3* [19,20] and *Foxp3* [21]. *Gata3* transcripts were not upregulated in tongue tissues from 4NQO-treated mice (Appendix A), whereas those of *Foxp3* were upregulated in these tissues at 16 weeks (Figure 7E). Moreover, we observed a moderately positive correlation between *Foxp3* and *Ccr4* transcript levels (*p* = 0.0113; Figure 7F) along with strong positive correlations between *Foxp3* and *Ccl17* (*p* = 0.0002; Figure 7G), *Ccl22* (*p* < 0.0001; Figure 7H), and *Il10* (*p* < 0.0001; Figure 7I). The levels of Treg-associated chemokines/cytokines and the Treg-associated chemokine receptor *Ccr4* were examined in dysplasia and SCC lesions (Appendix A). Although *Ccr4* was decreased in SCC, there was no significant decrease in *Ccl17*, *Ccl22*, and *Il10* levels in SCC. These results suggest that CCL17 and CCL22 recruit Foxp3^+^ IL-10-expressing Tregs during the 4NQO-induced development of premalignant lesions.

### 3.5 Infiltration of CD4^+^, CD8^+^, and Foxp3+ Cells in Carcinogen-Induced Mouse Tongue Tumorigenesis

To validate the infiltration of CD4^+^, CD8^+^, and Foxp3^+^ cells, we performed IHC analysis on paraffin-embedded specimens from tongue tissues from 4NQO-treated mice. Although only a few CD4^+^ cells were detected in the tongue tissue from control mice (Figure 8A), CD4^+^ cells were observed in the subepithelial lesion in hyperplasia (Figure 8B), and CD4^+^ cells were found along with the rete pegs of the epithelium (Figure 8B, insert). Intraepithelial localization of CD4^+^ cells was observed in mild dysplasia and SCC (Figure 8C, D). In agreement with the localization of CD4^+^ cells, although few cells were observed in the tongue tissue of control mice (Figure 9A), CD8^+^ cells were also found in the subepithelial and intraepithelial lesions in 4NQO-treated mice (Figure 9B–D). Similarly, although Foxp3^+^ cells were not observed in the tongue tissue of control mice (Figure 10A), infiltrated Foxp3^+^ cells localized in sub-epithelial and intraepithelial lesions. However, the number of infiltrated Foxp3^+^ cells tended to be lower than that of CD4^+^ and CD8^+^ cells (Figure 10B–D).

To determine the correlation between infiltrated T cells and transcript levels of Th1-associated chemokines and IFNγ, we performed Spearman’s rank correlation coefficient analysis (Figure 11). The number of infiltrated CD4^+^ cells was positively correlated with *Cxcl9* (*p* = 0.0054; Figure 11A), *Cxcl10* (*p* = 0.001; Figure 11B), and *Ifng* (*p* = 0.0002; Figure 11C) expression, and strong positive correlations were observed between the infiltrated CD8^+^ cells and transcript levels of Th1-associated chemokines (*Cxcl9* (*p* = 0.0028; Figure 11D) and *Cxcl10* (*p* = 0.0001; Figure 11E)), as well as *Ifng* (*p* < 0.0001; Figure 11F). Interestingly, a strong positive correlation between infiltration of CD8^+^ cells and *Il10* expression was observed (Appendix A).

Similarly, we analyzed correlations between infiltrating T cells and transcript levels of Th2/Treg-associated chemokines/cytokines (Figure 12). The number of infiltrated CD4^+^ cells positively correlated with *Ccl17* (*p* = 0.001; Figure 12A), *Cxcl22* (*p* = 0.002; Figure 12B), and *Il10* (*p* < 0.0001; Figure 12C) levels, and the number of infiltrated Foxp3^+^ cells positively correlated with *Ccl17* (*p* = 0.0059; Figure 12D), *Ccl22* (*p* = 0.0009; Figure 12E), and *Il10* levels (*p* = 0.0159; Figure 12F).

These results indicate that CD4^+^, CD8^+^, and Foxp3^+^ cells were simultaneously recruited through the expression of Th1- and Treg-associated chemokines in OPLs associated with 4NQO-induced mouse tongue tumorigenesis.

## 4. Discussion

Immune cell infiltration into local tissue where tumors develop is a key factor in the development and progression of oral cancer, as well as in antitumor activity. Although the immune phenotypes of infiltrating and circulating lymphocytes in OSCC have been extensively studied [6,22,23,24,25], chemokine expression responsible for immune cell infiltration through the progression of premalignant lesions to OSCC has not been fully elucidated [26,27,28,29]. In the present study, we examined the gene expression of chemokines/cytokines in the 4NQO-induced OPL and OSCC development.

We initially analyzed the gene expression of chemokines/cytokines using a PCR array and then validated results by qRT-PCR. The results demonstrate that genes for both Th1-associated chemokines (*Cxcl9, Cxcl10*, and *Ccl5*) and Th2/Treg-associated chemokines (*Ccl17* and *Ccl22*) were simultaneously expressed in 4NQO-induced murine OPLs, and as the tumor progressed to OSCC, the expression of Th1- and Th2/Treg-associated chemokine genes decreased. The Th1-type chemokines CXCL9 and CXCL10 are chemoattractants for activated T cells, including Th1 cells, CTLs, and NK cells, which preferentially express the chemokine receptor CXCR3 [16]. Consistent with Th1-associated chemokine gene expression, transcript levels of the chemokine receptor *Cxcr3* were upregulated in 4NQO-induced premalignant lesions, and a strong positive correlation was observed between *Cxcr3* level and *Cxcl9* and *Cxcl10* expression. Furthermore, strong positive correlations were observed between the number of infiltrated CD8^+^ cells and *Cxcl9*, *Cxcl10*, and *Ifng* levels. In addition, a strong positive correlation between infiltration of CD8^+^ cells and *Il10* expression was observed. Although the functional significance of the positive correlation between the CD8^+^ cells and *Il10* expression in 4NQO-induced tumorigenesis remains to be determined, IL-10 has been shown to participate in intratumoral tumor-specific cytotoxic CD8^+^ T cell infiltration and activation [30]. These results suggest that Th1/CTL-mediated antitumor immune responses are induced in premalignant lesions.

Interestingly, our results also indicate that transcript levels of Th2/Treg-associated chemokines *Ccl17* and *Ccl22* were increased in 4NQO-induced premalignant mouse tongue lesions at 16 weeks. Concomitant with the elevated expression of these chemokine genes, we also observed an increase in *Ccr4* expression. Although CCR4 is expressed on Th2 cells and Tregs [17], the following observations suggest that transcript levels of *Ccr4* could be derived from Tregs. First, transcript levels of *Foxp3* (a Treg marker [21]) but not *Gata3* (a Th2 marker) [19,20] were upregulated in OPLs. Second, IHC analysis revealed Foxp3^+^ cells in 4NQO-induced OPLs, and that the number of infiltrated Foxp3^+^ cells correlated with transcript levels of *Ccl17* or *CCl22*. These results suggest that CCR4^+^ Tregs are recruited in premalignant lesions by the chemokines CCL17 and CCL22. The elevated levels of *Ccr4* and *Foxp3* declined when the lesions progressed to advanced premalignant lesions at 28 weeks, along with a decrease in transcript levels of *Ccl17* and *Ccl22.* Taken together, these results suggest that, during the process of premalignant-lesion development, an antitumor immune response mediated by infiltrating Th1 cells/CTLs and Treg-mediated immune suppression appear to be triggered. As the premalignant lesions progress to cancerous lesions, both antitumor immunity and Treg-mediated immune suppression in the tumor lesions seemingly decline.

Our results agree with several previous studies on the immunological environment during the process of carcinogenesis in human OPLs and on rodent models using the carcinogen 4NQO [8,23,26]. We previously showed that human oral leukoplakia, a premalignant oral lesion, is a Th1-dominated tumor microenvironment that shows increased CXCL9 expression and infiltration of CXCR3^+^ and CCR5^+^ cells [8]. In a rodent model, cervical lymph node cells from 4NQO-induced premalignant and OSCC-bearing mice exhibited increased Th1 and CTL responses without concomitant changes in the Th2 response [22]. Similar to the present study findings, Woodford et al. [23] reported that levels of the Th1-type cytokine IFNγ were increased in 4NQO-induced mouse premalignant lesions, whereas these levels were reduced in established OSCC lesions. Additionally, Miki et al. [28] reported that elevated *Foxp3* expression in OPLs decreased along with tumor progression. Furthermore, other studies indicated that decreased numbers of infiltrated T cells in the tumor microenvironment during progression to OSCC result in increased levels of T cells in regional lymph nodes, which correlates with tumor development [22,24]. In this regard, De Costa et al. [22] reported that numbers of Th1 cells and CTLs, as well as Tregs, are increased in the cervical lymph nodes of OSCC-bearing mice; further, the proliferative capacity of CD4^+^ T cells and the suppressive capacity of Treg were progressively decreased throughout OSCC development, and the Th1 cells and CTL were found to express the exhaustion marker, programmed death-1 (PD-1). These findings suggest that activated Th1 cells and CTLs, as well as Tregs, are recruited in the tumor microenvironment of OPLs and exert antitumor immune responses against dysplastic cells, as well as concomitant immunosuppression, to dampen effective Th1/CTL-mediated responses. As the tumor progresses to OSCC during a chronic immune response, tumor-specific T cells with a hyporesponsive phenotype reside in regional lymph due to the decreased Th1- and Treg-associated chemokine expression in the local tumor microenvironment.

The molecular mechanisms by which 4NQO induces Th1- and Treg-associated chemokine gene expression in mouse tongue tissues remain unknown. However, 4NQO mimics the carcinogenic effects of tobacco and can induce intracellular oxidative stress, resulting in the generation of oxidatively damaged DNA bases, such as 8-oxo-7,8-dihydroguanine (8-oxoG) [31,32]. The generated 8-oxoG is then excised and released as a free base by the DNA base-excision repair pathway [31]. Accumulation of intracellular 8-oxoG induces activation of Ras and small GTPases, inducing the expression of proinflammatory chemokine/cytokine genes, including *Tnfa, Il1*, and *Ccl3* [32,33]. Although activation of Ras and small GTPases occurs within minutes after 8-oxoG exposure, the cytokine/chemokine gene expression observed in 4NQO-induced tongue tumorigenesis persisted for months. These results suggest that continuous tongue tissue exposure to 4NQO for 16 weeks triggers Ras signaling pathway activation by oxidative stress. Furthermore, the expression of proinflammatory cytokines, such as *Tnfa* and *Il1b*, induced by 8-oxoG, also induces a wide variety of chemokines/cytokines. In this context, *Il1b* reportedly acts as a node gene during 4NQO-induced carcinogenesis [34]. In the present study, PCR array analysis showed high levels of *Il1b* expression in mouse tongue tissues at 28 weeks after 4NQO treatment. However, the status of 4NQO-mediated *Il1b* expression as a key process for triggering cytokine/chemokine gene expression in mouse tongue tissues has not yet been investigated.

OPLs precede OSCC, and lower T cell infiltration in OPLs is associated with OSCC [5,6,7]. However, not all OPLs develop into OSCC, and some may even regress [4], suggesting that T cell-mediated adaptive immunity protects against malignant transformation. Therefore, analysis of the expression of Th1- and Treg-associated chemokine/cytokine levels in biopsy specimens from patients with OPLs could predict patient prognosis. In addition, targeting of CCR4 by small-molecule antagonists and monoclonal antibodies is reportedly an effective immunotherapeutic strategy [17,35,36]. Thus, preclinical models of OSCC using 4NQO are useful to test the therapeutic efficacy of targeting Treg-associated chemokines.

Although we only focused on the expression of Th1- and Treg-associated chemokine genes and the infiltration of CD4^+^, CD8^+^, and Foxp3^+^ cells in 4NQO-induced mouse tongue tumorigenesis, other immune cells, including myeloid-derived suppressor cells [37], eosinophils [38], Th17 cells [23], tumor-associated macrophages [39], and the chemokines responsible for the infiltration of these cells also reportedly participate in tumorigenesis. Furthermore, the tumor microenvironment comprises infiltrated immune cells, as well as stromal cells, such as cancer-associated fibroblasts [40]. The mechanisms by which immune cells and stromal cells in the premalignant tumor microenvironment orchestrate the expression of chemokines/cytokines that promote OSCC progression remain to be determined. Additionally, whether the relative expression levels of Th1- and Treg-associated chemokines/cytokines in the tumor microenvironment contribute to malignant transformation from premalignant lesions warrant further investigation. Thus, further studies are needed to elucidate the mechanisms involved in the 4NQO-induced expression of Th1- and Treg-associated chemokine genes during the development of tongue tumorigenesis.

## 5. Conclusions

This study demonstrated that the genes for both Th1- and Treg-associated chemokines were simultaneously expressed in 4NQO-induced murine OPLs. Additionally, results indicate that CD4^+^, CD8^+^, and Foxp3^+^ cells were simultaneously recruited through the expression of Th1- and Treg-associated chemokines in premalignant lesions. These findings indicate that Th1/CTL-mediated antitumor immune response and Foxp3^+^ Treg-mediated immunosuppression appear to be simultaneously initiated during the development of OPLs. Although the mechanisms involved in the simultaneous expression of the Th1- and Treg-associated chemokine genes in OPLs remain to be determined, downregulation of Treg-associated chemokine expression and/or intervention into chemokine receptor-ligand interactions might prevent OSCC development. Further studies are essential to elucidate the role of the simultaneous expression of Th1- and Treg-associated chemokines in the development of OSCC and test the therapeutic efficacy of targeting Treg-associated chemokines in preclinical models using 4NQO.

## Figures and Tables

**Figure 1 cancers-13-01835-f001:**
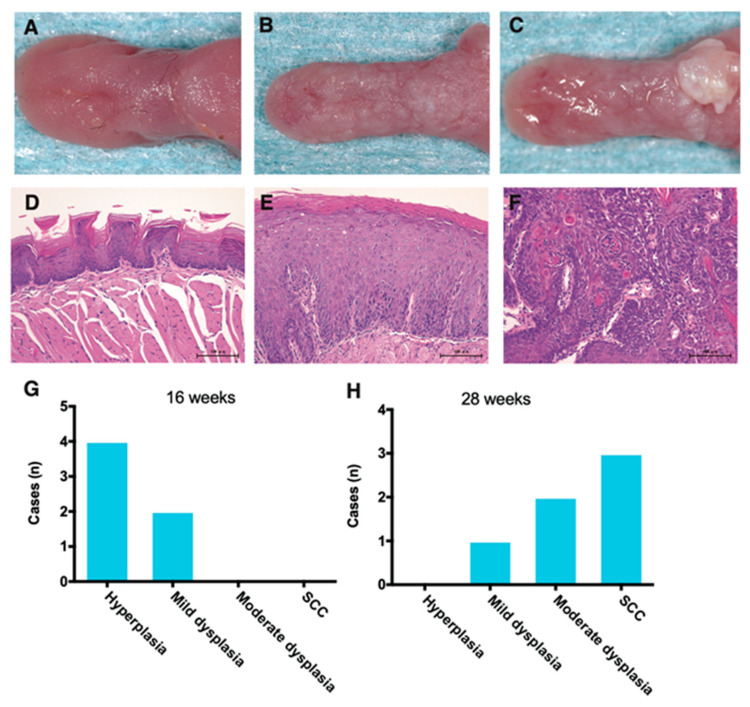
Gross lesions and hematoxylin and eosin (H-E) staining of mouse tongue tumors induced by the carcinogen 4NQO. Mice were treated with 4NQO in drinking water (50 μg/mL) for 16 weeks and then changed to regular drinking water for another 12 weeks, and tongue tissues were dissected at 16 and 28 weeks. Gross lesions of the normal tongue from mice treated with the vehicle for 28 weeks (**A**). A dysplastic lesion in a 4NQO-treated mouse at 28 weeks (**B**). A large invasive squamous cell carcinoma (SCC) from a 4NQO-treated mouse (28 weeks) (**C**). Representative images of H-E staining of paraffin-embedded tongue tissue sections of normal squamous epithelium (**D**); moderate dysplasia lesions (**E**); and SCC lesions (**F**) (D-F: original magnification, 40×; scale bar = 100 μm). Histopathological grade of 4NQO-indued mouse tongue tumors at 16 weeks (**G**); and 28 weeks (**H**).

**Figure 2 cancers-13-01835-f002:**
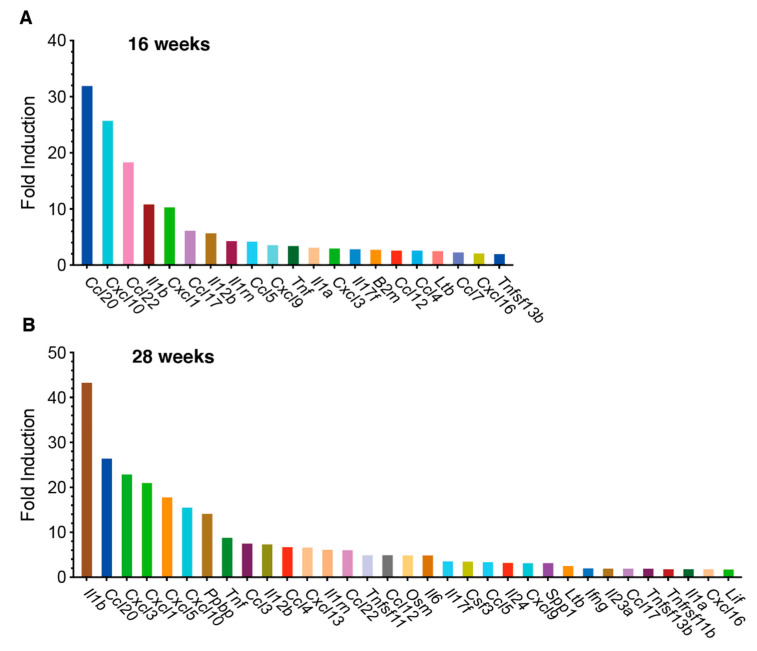
PCR array analysis of cytokine and chemokine gene expression in tongue tissues from 4NQO-treated mice. Cytokine and chemokine gene expression levels in tongue tissues from 4NQO- (*n* = 6) and vehicle-treated (*n* = 6) mice at 16 weeks (**A**) and 28 weeks (**B**), respectively, were analyzed using the RT^2^ Profiler PCR Array. Changes in gene expression are expressed as fold changes of levels in 4NQO-treated mice relative to the vehicle. Changes in gene expression >2-fold (*p* < 0.05) are shown in the figure.

**Figure 3 cancers-13-01835-f003:**
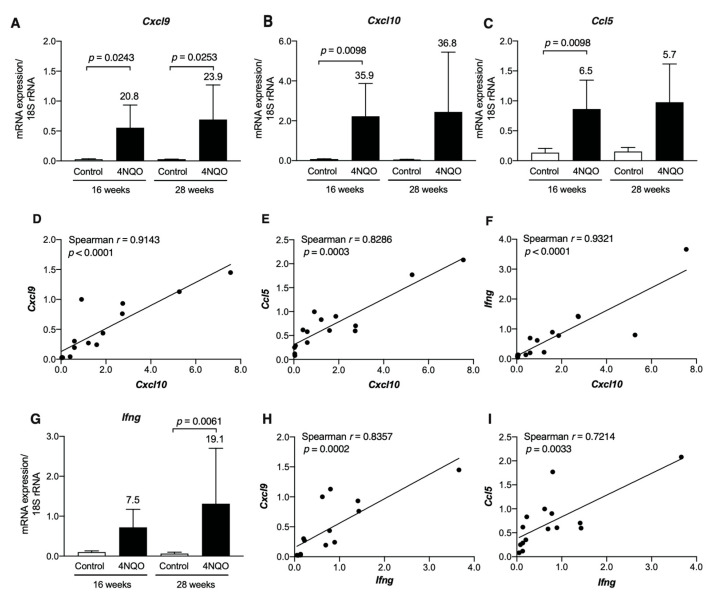
Increases in transcript levels of Th1-associated chemokines/cytokines in tongue tissues of 4NQO-treated mice. mRNA levels in tongue tissues of mice treated with vehicle (control) or 4NQO were assessed at the indicated time points by qRT-PCR: *Cxcl9* (**A**); *Cxcl10* (**B**); *Ccl5* (**C**); and *Ifng* (**G**). The relative mRNA levels were normalized to 18S rRNA levels. Each column and bar represent the mean ± SD (*n* = 6). Statistically significant differences in the mRNA expression of 4NQO-treated mice relative to control mice are indicated. The fold induction of mRNA expression in tongue tissues from mice treated with 4NQO relative to that in control mice is shown above the column. Correlations between transcript levels of *Cxcl10* and *Cxcl9* (**D**), *Ccl5* (**E**), and *Ifng* (**F**) and between transcript levels of *Ifng* and *Cxcl9* (**H**) and *Ccl5* (**I**) in tongue tissues were determined using Spearman’s rank correlation coefficient analysis.

**Figure 4 cancers-13-01835-f004:**
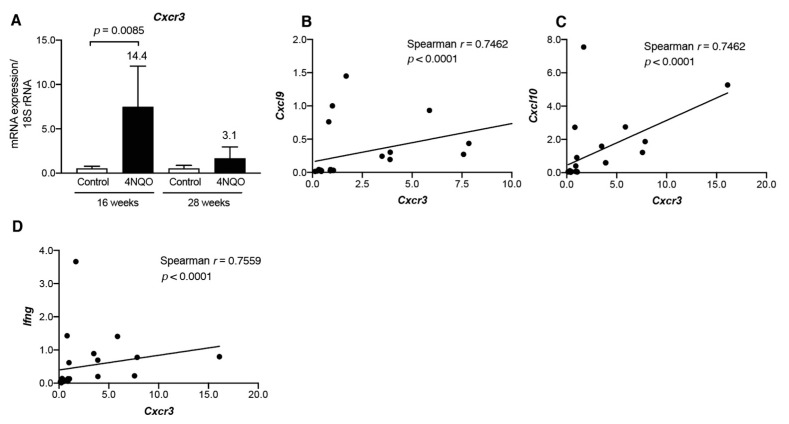
*Cxcr3* transcript levels in tongue tissues from 4NQO-treated mice. mRNA levels of *Cxcr3* (**A**) in the tongue tissues of mice treated with vehicle (control) or 4NQO were assessed at the indicated time points by qRT-PCR. Relative mRNA levels were normalized to 18S rRNA. Each column and bar represent the mean ± SD (*n* = 6). Statistically significant differences in the mRNA expression of 4NQO-treated mice relative to control mice are indicated. Fold induction of mRNA expression in tongue tissue from mice treated with 4NQO relative to that in control mice is shown above the column. Correlations between transcript levels of *Cxcr3* and *Cxcl9* (**B**), *Cxcl10* (**C**), and *Ifng* (**D**) in tongue tissue were determined using Spearman’s rank correlation coefficient analysis.

**Figure 5 cancers-13-01835-f005:**
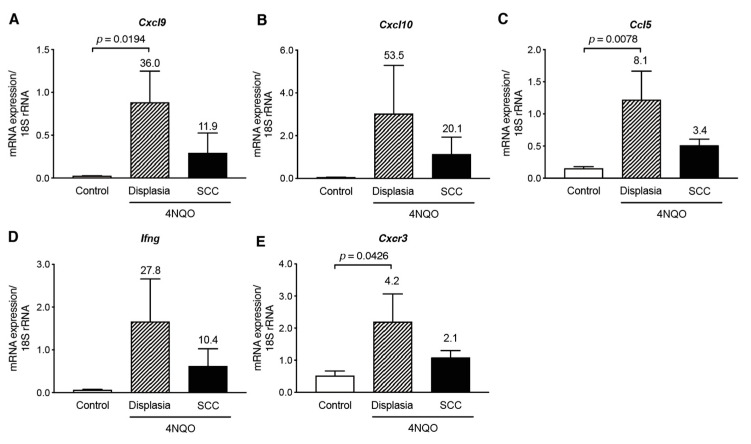
Decreases in the transcript levels of Th1-associated chemokines/cytokines and *Cxcr3* in SCC tongue tissue at 28 weeks after 4NQO treatment. mRNA levels in tongue tissue of control mice or dysplastic or SCC lesions were assessed by qRT-PCR: *Cxcl9* (**A**); *Cxcl10* (**B**); *Ccl5* (**C**); *Ifng* (**D**); and *Cxcr3* (**E**). Relative mRNA levels were normalized to 18S rRNA. Each column and bar represent the mean and SD (*n* = 6) for the control, dysplasia, and SCC mice (*n* = 3 each). Statistically significant differences in mRNA expression in 4NQO-treated mice relative to control mice are indicated. Fold induction of mRNA expression in tongue tissues from 4NQO-treated mice relative to that in control mice, is shown above the column.

**Figure 6 cancers-13-01835-f006:**
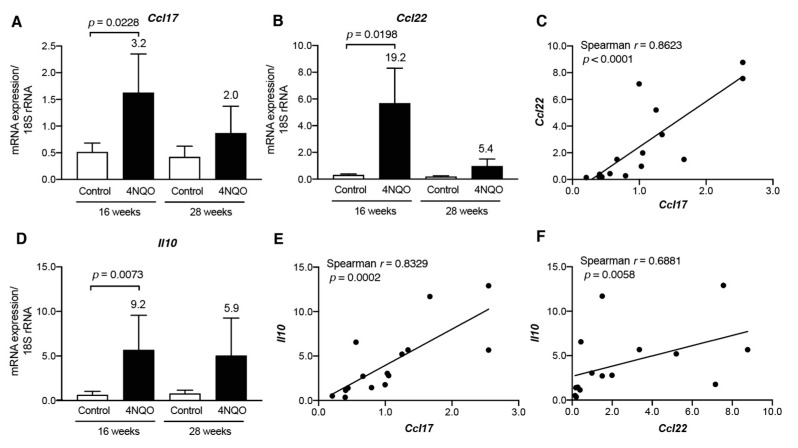
Increased transcript levels of Th2/Treg-associated chemokines/cytokines in tongue tissues from 4NQO-treated mice. mRNA levels in tongue tissues of mice treated with vehicle (control) or 4NQO were assessed at the indicated time points by qRT-PCR: *Ccl17* (**A**); *Ccl22* (**B**); and *Il10* (**D**). Relative mRNA levels were normalized to those of 18S rRNA. Each column and bar represent the mean ± SD (*n* = 6). Statistically significant differences in mRNA expression in 4NQO-treated mice relative to control mice are indicated. Fold induction of mRNA expression in tongue tissues from mice treated with 4NQO relative to that in control mice is shown above the column. Correlations between transcript levels of *Ccl17* and *Ccl22* (**C**) and *Il10* (**E**) and between *Ccl22* and *Il10* (**F**) were determined using Spearman’s rank correlation coefficient analysis.

**Figure 7 cancers-13-01835-f007:**
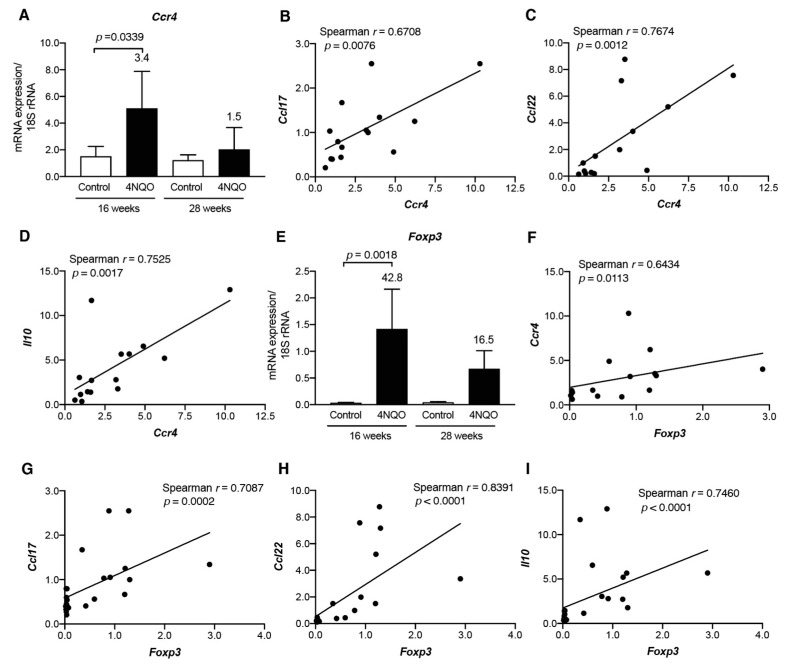
Increased transcript levels of *Ccr4* and *Foxp3* in tongue tissues from 4NQO-treated mice. mRNA levels of *Ccr4* (**A**) and *Foxp3* (**E**) in tongue tissues of mice treated with vehicle (control) or 4NQO were assessed at the indicated time points by qRT-PCR. Relative mRNA levels were normalized to those of 18S rRNA. Each column and bar represent the mean ± SD (*n* = 6). Fold induction of mRNA expression in tongue tissues from mice treated with 4NQO relative to that in control mice is shown above the column. Correlations between the transcript levels of *Ccr4* and *Ccl17* (**B**), *Ccl22* (**C**), and *Il10* (**D**) and between *Foxp3* and *Ccr4* (**F**) *Ccl17* (**G**), *Ccl22* (**H**), and *Il10* (**I**) were determined using Spearman’s rank correlation coefficient analysis.

**Figure 8 cancers-13-01835-f008:**
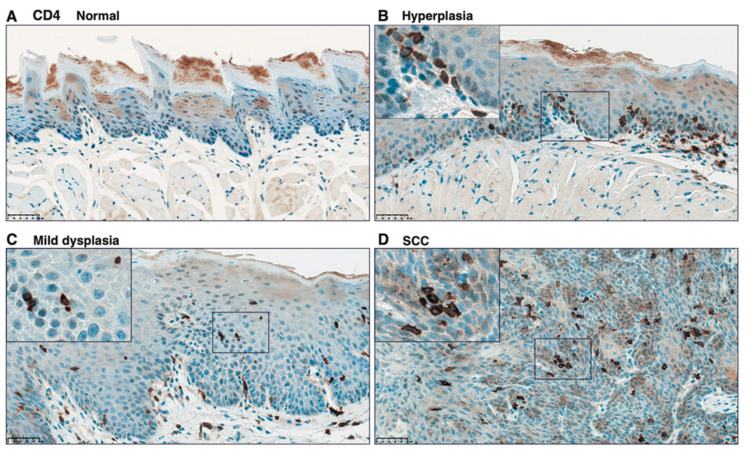
Immunohistochemical analysis of CD4^+^ cells in tongue tissue from 4NQO-treated mice. Representative CD4 staining of tongue tissue from mice treated with vehicle (control) (**A**); 4NQO at 16 weeks (**B**,**C**); and 4NQO at 28 weeks, S (**D**). Pathologic determinations using H-E staining are shown. Original images were captured at 400×, and inset images were captured at 800×. Scale bar = 50 μm.

**Figure 9 cancers-13-01835-f009:**
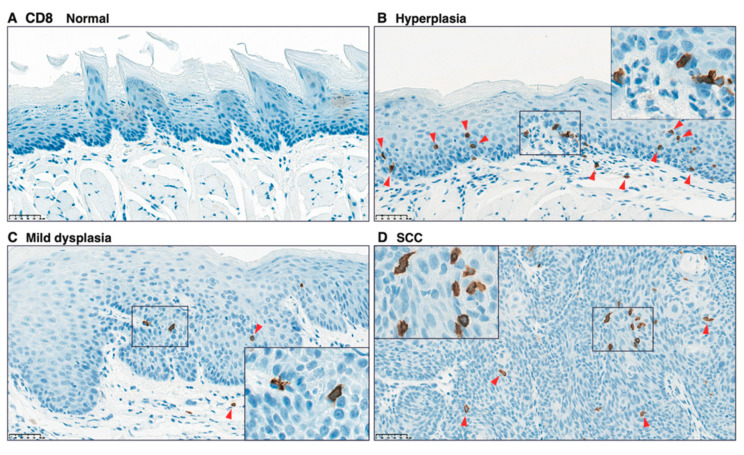
Immunohistochemical analysis of CD8^+^ cells in tongue tissues from 4NQO-treated mice. Representative CD8 staining of tongue tissue from mice treated with vehicle (control) (**A**); 4NQO at 16 weeks (**B**,**C**); and 4NQO at 28 weeks (**D**). Pathologic determinations using H-E staining are shown. Original images were captured at 400×, and inset images were captured at 800×. Arrowheads indicate CD8^+^ cells. Scale bar = 50 μm.

**Figure 10 cancers-13-01835-f010:**
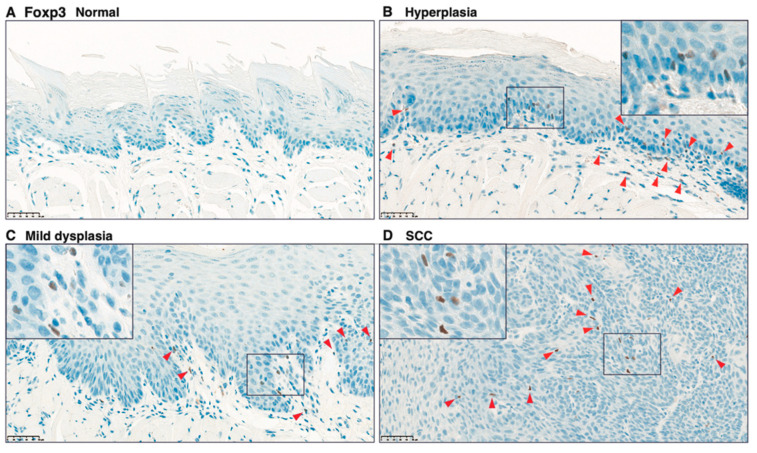
Immunohistochemical analysis of Foxp3^+^ cells in tongue tissues from 4NQO-treated mice. Representative Foxp3 staining of tongue tissues from mice treated with vehicle (control) (**A**); 4NQO at 16 weeks (**B**,**C**); and 4NQO at 28 weeks (**D**). Pathologic determinations using H-E staining are shown. Original images were captured at 400×, and inset images were captured at 800×. Arrowheads indicate Foxp3^+^ cells. Scale bar = 50 μm.

**Figure 11 cancers-13-01835-f011:**
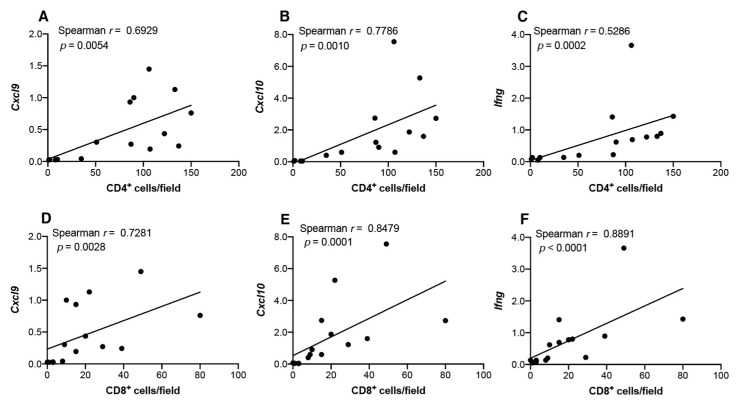
Correlation between transcript levels of Th1-associated chemokines/cytokines and the infiltration of CD4^+^ or CD8^+^ cells in tongue tissues from 4NQO-treated mice. mRNA levels of *Cxcl9* (**A**,**D**), *Cxcl10* (**B**,**E**), and *Ifng* (**C**,**F**) in tongue tissues from each mouse were assessed for correlations with infiltrated CD4^+^ (**A**–**C**) or CD8^+^ cells (**D**–**F**) via Spearman’s rank correlation coefficient analysis.

**Figure 12 cancers-13-01835-f012:**
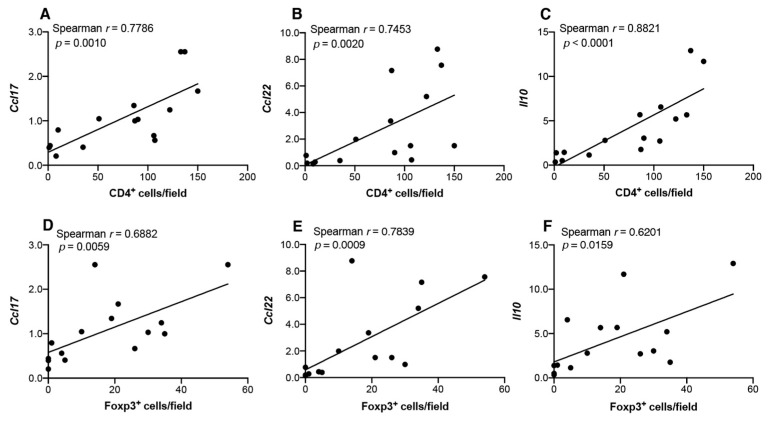
Correlation between transcript levels of Th2/Treg-associated chemokines/cytokines and the infiltration of CD4^+^ or Foxp3^+^ cells in tongue tissues from 4NQO-treated mice. mRNA levels of *Ccl17* (**A**,**D**), *Ccl22* (**B**,**E**), and *Il10* (**C**,**F**) in tongue tissues from each mouse were assessed for correlations with infiltrated CD4^+^ (**A**–**C**) or Foxp3^+^ cells (**D**–**F**) via Spearman’s rank correlation coefficient analysis.

## Data Availability

The data presented in this study are available in this article and the Appendix A.

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
