# Peer review of "Simultaneous Expression of Th1- and Treg-Associated Chemokine Genes and CD4^+^, CD8^+^, and Foxp3^+^ Cells in the Premalignant Lesions of 4NQO-Induced Mouse Tongue Tumorigenesis"

_cancers, 2021, doi:10.3390/cancers13081835_

Round 1

Reviewer 1 Report

The genes for both Th1- and Treg-associated chemokines were found to be expressed simultaneously in 4NQO-induced murine OPLs in this study. Furthermore, the findings showed that CD4+, CD8-positive, and Foxp3+ cells were all presented simultaneously. These results suggest that antitumor immune responses mediated by Th1/CTL and Foxp3+ Treg-mediated immunosuppression appear to start at the same time in the developing of OPLs.

The aim in this excellent paper is stated clear. The authors stated clearly what study found and how they did it. The title is informative and relevant.

The references are relevant and recent. Appropriate and key studies are included.

The study methods are valid and reliable. There are enough details provided in order to replicate the study.

The data is presented in an appropriate way. Results are discussed from different angles and placed into context without being overinterpreted.

The conclusions answer the aim of the study. The conclusions are supported by references and own results.

Specific comments on weaknesses of the article and what could be improved:

Major points - none

Minor points

  1. Please, state the limitations of the study
  2. Could you please discuss the clinical implications of the results?

Author Response

Reply to Reviewer #1

  1. Please, state the limitations of the study.

The limitation of this study is that we only focused on the expression of Th1- and Treg-associated chemokines and the infiltration of CD4+, CD8+, and Foxp3+ cells in 4NQO-induced mouse tongue tumorigenesis. We had already stated the concerns in the Discussion of the original manuscript (lines from 487 to 500). 

  1. Could you please discuss the clinical implications of the results?

 We have discussed and incorporated the clinical implications and perspectives in the revised manuscript as follows:

Lines 478 - 486:

 OSCC is preceded by OPLs and lower T cell infiltration in OPLs is associated with OSCC [5-7]. However, not all OPLs develop into OSCC, and some may even regress [4], suggesting that T cell-mediated adaptive immunity protects against malignant trans-formation. Therefore, analysis of the expression of Th1- and Treg-associated chemokine/cytokine levels in biopsy specimens from patients with OPLs could predict patient prognosis. In addition, several reports have focused on the targeting of CCR4 by small molecule antagonists and monoclonal antibodies as an effective immunotherapeutic strategy [17, 35, 36]. Thus, preclinical models of OSCC using 4NQO is useful to test the therapeutic efficacy of targeting Treg-associated chemokines.

For your reference:

  1. Yoshie, O.; Matsushima, K. CCR4 and its ligands: from bench to bedside. Int Immunol 2015, 27, 11-20, doi:10.1093/intimm/dxu079.
  2. Solari, R.; Pease, J.E. Targeting chemokine receptors in disease--a case study of CCR4. Eur J Pharmacol 2015, 763, 169-177, doi:10.1016/j.ejphar.2015.05.018.
  3. Sun, W.; Li, W.J.; Wei, F.Q.; Wong, T.S.; Lei, W.B.; Zhu, X.L.; Li, J.; Wen, W.P. Blockade of MCP-1/CCR4 signaling-induced recruitment of activated regulatory cells evokes an antitumor immune response in head and neck squamous cell carcinoma. Oncotarget 2016, 7, 37714-37727, doi:10.18632/oncotarget.9265.

Reviewer 2 Report

It is overall a very intesresting study and generally well-written article. I have just 6 minor comments - below:

1. Please clearly state dilution for each antibody used for IHC
2. Have you considered checking also genes associated with MDSC cells e.g. IDO, arginase, TGF-beta? I would expect a significant upregulation, especially in SCC - the oposite trend to the one observed for Th1-related. 
3. "demonstrated  a  high  positive  correlation" - strong?
4. Could the authors provide data for IL-10 expression for dysplasia and SCC separately? I would expect an increase in both, but much higher in SCC. That could be partial explanation for Th1-related decrease
5. Please also test correlation between CD8 and CD4 infilatrion on one side and IL-10 expression on the other
6. "we examined the gene-expression kinetics" - kinetics in this case seem to be a bit of overstatement. There were only two time points.

Author Response

Reply to Reviewer #2

  1. Please clearly state dilution for each antibody used for IHC.

We have added the dilution for each antibody in the IHC staining section (line 155 – 159) as follows:

Lines 155 – 160:

The tissue sections were then incubated with rabbit monoclonal anti-mouse CD4 anti-body (1:100 dilution, Cat#25229, Cell Signaling Technology, Danvers, MA, USA), rabbit monoclonal anti-mouse CD8a antibody (1:200 dilution, Cat#98941, Cell Signaling Technology), or rabbit monoclonal anti-mouse Foxp3 antibody (1:100 dilution, Cat#12653, Cell Signaling Technology) at 1:100 and 25°C in a humidified chamber for 60 min.

  1. Have you considered checking also genes associated with MDSC cells e.g. IDO, arginase, TGF-beta? I would expect a significant upregulation, especially in SCC - the opposite trend to the one observed for Th1-related.

We initially examined chemokine/cytokine gene-expression profiles using PCR array analysis. In this assay, transcript levels of TGF-beta (Tgfb2) were not up-regulated in the tongue tissue of 4NQO-treated mice. Although we did not examine IDO expression by using qRT-PCR, transcript levels of Agr1 were upregulated in the tongue tissue of 4NQO-treated mice at 28 weeks. However, there were no significant differences in the transcript levels between dysplasia and SCC. We are currently analyzing other genes associated with MDSC and TAMs.

  1. "demonstrated a high positive correlation" - strong?

We have corrected the phrase “demonstrated a high positive correlation” to “demonstrated a strong positive correlation” (line 292).

  1. Could the authors provide data for IL-10 expression for dysplasia and SCC separately? I would expect an increase in both, but much higher in SCC. That could be partial explanation for Th1-related decrease.

We had already provided the data regarding IL-10 expression for dysplasia and SCC in the Supplemental Figure S3, panel D (original MS, line 319 – 320). As you indicated, transcript levels of Il10 were upregulated in both. Although there were no statistical differences in the transcript levels between dysplasia and SCC, the levels of Il10 tended to be upregulated in SCC. Based on your suggestion, we have modified the statement regarding Treg-associated chemokine/cytokine expression in dysplasia and SCC lesions as follows:

Lines 318 - 322:

The levels of Treg-associated chemokines/cytokines and the Treg-associated chemokine receptor Ccr4 were examined in dysplasia and SCC lesions (Supplementary Figure S3). Although Ccr4 was decreased in SCC, there was no significant decrease in the levels of Ccl17, Ccl22 and Il10 in SCC.

  1. Please also test correlation between CD8 and CD4 infiltration on one side and IL-10 expression on the other.

We had also provided the data regarding the correlation between CD4+ and IL-10 expression (Figure 12C), (original MS, line 377). The data shows that there is a strong positive correlation (p < 0.0001) between infiltration of CD4+ cells and Il10 expression (original MS, line 379).

Based on the suggestion of the reviewer, we have analyzed the correlation between infiltration of CD8+ cells and Il10expression and found a strong positive correlation between infiltration of CD8+ cells and Il10 expression. Although the functional significance of the positive correlation between the infiltration of CD8+ cells and Il10 expression in 4NQO-induced tumorigenesis remains to be determined, IL-10 has been shown to participate in infiltration and activation of intratumoral tumor-specific cytotoxic CD8+ T cells [30]. We have provided the data for the correlation between infiltration of CD8+ cells and Il10 expression as Supplementary Figure S4 and added information regarding the correlation as follows:

Lines 366 – 368:

Interestingly, a strong positive correlation between infiltration of CD8+ cells and Il10 expression was observed (Supplementary Figure S4).

Lines 410 - 415:

In addition, a strong positive correlation between infiltration of CD8+ cells and Il10 expression was observed. Although the functional significance of the positive correlation between the CD8+ cells and Il10 expression in 4NQO-induced tumorigenesis remains to be determined, IL-10 has been shown to participate in infiltration and activation of intratumoral tumor-specific cytotoxic CD8+ T cells [30]. These results suggested that Th1/CTL-mediated antitumor immune responses are induced in premalignant lesions.

For your reference:

  1. Mumm, J.B.; Emmerich, J.; Zhang, X.; Chan, I.; Wu, L.; Mauze, S.; Blaisdell, S.; Basham, B.; Dai, J.; Grein, J., et al. IL-10 elicits IFNgamma-dependent tumor immune surveillance. Cancer Cell 2011, 20, 781-796, doi:10.1016/j.ccr.2011.11.003.

  1. "we examined the gene-expression kinetics" - kinetics in this case seem to be a bit of overstatement. There were only two time points.

We have deleted “kinetics” (line 396). 

Reviewer 3 Report

The finding reported by Yamaguchi et al provides a very significant insight regarding the role of simultaneous expression of Th1- and Treg-associated chemokine genes in the development of oral squamous cell carcinoma. The study is very well carried out and the results are interesting. The outcome of this study will facilitate the process of cancer drug design and development. I recommend the acceptance of this manuscript in the present form.  

Author Response

Thank you very much for your evaluation.